# A Comparison of Clinical Manifestations and Outcomes between Acute Sporadic Hepatitis A and Hepatitis E Infections in Thailand

**DOI:** 10.3390/v15091888

**Published:** 2023-09-07

**Authors:** Sirajuk Khongviwatsathien, Wajana Thaweerat, Thanapat Atthakitmongkol, Watcharasak Chotiyaputta, Tawesak Tanwandee

**Affiliations:** Division of Gastroenterology, Department of Medicine, Faculty of Medicine Siriraj Hospital, Mahidol University, Bangkok 10700, Thailandwajana.tha@gmail.com (W.T.); watcharasak.cho@mahidol.ac.th (W.C.)

**Keywords:** clinical presentation, hepatitis A, hepatitis E, outcome, Thailand

## Abstract

Hepatitis A virus (HAV) and hepatitis E virus (HEV) infections often present as acute hepatitis with prodromal symptoms. These infections, transmitted via the oral–enteral route, constitute significant public health challenges, particularly in developing countries with subpar sanitary systems. The aim of the study was to describe the clinical manifestations, laboratory findings, and outcomes of hepatitis A and hepatitis E infections in Thailand. We conducted a retrospective chart review and analysis of 152 patients diagnosed with acute hepatitis A or hepatitis E from January 2007 to August 2018 at Siriraj Hospital. The hepatitis E cohort was older with a greater prevalence of comorbidities (hypertension, diabetes mellitus, chronic kidney disease, chronic hepatitis B, and post-kidney transplantation status) than the hepatitis A cohort. While the majority of hepatitis A patients presented with fever (98%) and jaundice (96%), these symptoms were less pronounced in hepatitis E patients. Furthermore, hepatitis A patients exhibited significantly higher aminotransferase and total bilirubin levels. However, clinical outcomes, such as hospitalization rates, progression to acute liver failure, and mortality, were comparable across both groups. In conclusion, although the clinical manifestations of hepatitis A and hepatitis E were similar, fever and jaundice were more prevalent and aminotransferase and bilirubin levels were higher in the HAV-infected group.

## 1. Introduction

Viral hepatitis remains a significant global health concern [1,2,3]. Hepatitis A virus (HAV) and hepatitis E virus (HEV) are among the most prevalent causes of acute hepatitis. The viruses affect millions of children and adults annually [1,2,3,4], especially in developing countries due to unfavorable socioeconomic conditions and inadequate sanitary systems [1,5,6,7,8]. Both HAV and HEV are primarily transmitted through the fecal–oral route as enterically transmitted hepatotropic viruses, often via contaminated water and food [3,4,5,6,9].

HAV, a single-stranded RNA virus, is classified within the Picornaviridae family and the *Hepatovirus* genus [6]. Predominantly, HAV transmission occurs from person to person [2,4]. It is estimated that 1.4 million individuals globally become infected each year [2], with approximately 11–22% of infected patients requiring hospitalization [10]. Clinical manifestations of acute HAV infection range from asymptomatic infection to acute liver failure without progression to chronic hepatitis [6,8,11]. Symptomatic illness occurs in approximately 76–97% of adult HAV infections [11,12], yet over 70% of pediatric patients remain asymptomatic or mildly symptomatic [6,11]. Anti-HAV seroprevalence rates in Thailand in the 1970s indicated that at least half of children aged 5–10 years and 100% of 30-year-olds were infected. However, by the 2000s, these rates had significantly declined, with a lower seroprevalence rate observed in children (less than 5% of 10-year-olds) and only 20–30% of young adults [13].

HEV, a small nonenveloped virus with a single-stranded RNA genome, displays two distinct epidemiological patterns [14]. In developing countries with high endemicity, HEV genotype 1 or 2 is commonly associated with high mortality rates among pregnant women. In contrast, in areas of lower endemicity in developed countries and some developing countries, HEV genotype 3 or 4 is usually linked with autochthonous hepatitis E infections [14,15,16]. In Thailand, HEV infection has been predominantly attributed to genotype 3 and linked to zoonotic transmission from swine [17,18,19], unlike neighboring countries where genotype 1 predominates [20].

Both HAV and HEV infections typically present similar clinical manifestations of acute hepatitis, including fever, malaise, nausea, vomiting, dark urine, jaundice, and abdominal discomfort [6,9,12,14,15,21,22,23]. However, we are unaware of any publications that have reported on these infections in Thailand. Consequently, we investigated the clinical patterns of acute sporadic hepatitis A and E to identify differences in clinical manifestations and outcomes between these viral hepatitis infections.

## 2. Materials and Methods

### 2.1. Study Patients

This study retrospectively reviewed all patients diagnosed with hepatitis A and hepatitis E at Siriraj Hospital, Bangkok, Thailand, from January 2007 to August 2018. Diagnoses were based on the International Classification of Diseases (ICD), 10th version, Clinical Modification (ICD-10-CM) diagnostic codes B15 and B17.2. Hepatitis A diagnoses were confirmed through positive anti-HAV immunoglobulin M detection [24]. Hepatitis E diagnoses were confirmed via positive anti-HEV immunoglobulin M and HEV PCR results. However, due to resource limitations, some patients in the study diagnosed with hepatitis E only showed positive anti-HEV immunoglobulin M [25]. The study protocol was authorized by the Siriraj Institutional Review Board (COA approval number Si 657/2017).

### 2.2. Medical Data Collection

Patient medical data were retrieved from electronic medical records. Collected demographic data comprised the age at the time of infection and sex. Records were also made of patient comorbidities, for example, hypertension, diabetes mellitus, dyslipidemia, chronic kidney disease, malignancy, liver status (chronic hepatitis or cirrhosis), and post-organ transplantation. Evaluations were made of risk factors such as the consumption of undercooked food, a history of blood transfusion, and intravenous drug use. Data were collected on clinical history, initial physical examination findings, and laboratory test results (alanine aminotransferase (ALT), aspartate aminotransferase (AST), alkaline phosphatase, total bilirubin, albumin, and international normalized ratio) during the clinical course of the disease. Clinical progression was also noted.

### 2.3. Serology Test

Blood samples were tested for anti-HAV immunoglobulin M detection using ARCHITECT HAVAb-IgM (Abbott Laboratories, Wiesbaden, Germany) and anti-HEV immunoglobulin M detection using Anti-Hepatitis E Virus (HEV) IgM ELISA (EUROIMMUN, Lübeck, Germany), following the manufacturer’s instructions.

The detection principle of anti-HAV immunoglobulin is a chemiluminescent microparticle immunoassay (CMIA) for the qualitative detection of IgM antibody to hepatitis A virus in sera. The cut-off value of >1.20 is regarded as positive, while those value <0.80 is regarded as negative, and the values between 0.80 and 1.20 are considered borderline. The sensitivity and specificity of HAV IgM following the manufacturer’s instructions were 98.58% and ≥99.0%, respectively. However, the post-marketing data revealed sensitivity and specificity of HAV IgM were 78.2–89.3% and 100%, respectively [26].

The detection principle of anti-HEV immunoglobulin M is an indirect enzyme-linked immunosorbent assay (ELISA) based on the binding of HEV antibodies in sera to HEV recombinant antigens (genotypes 1 and 3). The cut-off value of ≥1.1 is regarded as positive, while those value <0.8 is regarded as negative, and the other values are considered borderline. The sensitivity and specificity of HEV IgM following the manufacturer’s information were 100% and 100%, respectively. However, subsequent post-marketing data indicated a sensitivity range of 24% to 61.5% and a consistent specificity of 100% for HEV IgM [27].

### 2.4. Statistical Analysis

Comorbidities and risk factors are presented as percentage values. Laboratory test results are summarized as median values, including initial, peak, and resolution levels. Continuous data were compared using the Student’s t test and are given as the means ± standard deviations for normally distributed data. The Mann–Whitney U test was used for nonnormally distributed data, and the results are reported as medians and interquartile ranges. Categorical data were compared using Fisher’s exact test, with results shown as percentages or as numbers and percentages. Clinical predictors to differentiate hepatitis E from hepatitis A infections were analyzed by univariable logistic regression analysis, followed by multivariable logistic regression analysis performed with a forward stepwise model. A *p* value less than 0.05 was considered statistically significant for all tests. All statistical calculations were performed using PASW Statistics, version 18 (SPSS Inc., Chicago, IL, USA).

## 3. Results

### 3.1. Baseline Characteristics

Between 2007 and 2018, 100 patients were diagnosed with hepatitis A and 52 with hepatitis E at Siriraj Hospital, Mahidol University. There was no hepatitis A and hepatitis E co-infection in this study. The diagnosis was confirmed through HEV PCR only in 14 patients (26.9%) in the hepatitis E group. The mean age was 37.0 ± 15.8 years, with hepatitis E patients generally older than hepatitis A patients. Both hepatitis groups were slightly male-dominant. Patients diagnosed with hepatitis E had a higher prevalence of comorbid conditions, such as hypertension, diabetes mellitus, chronic kidney disease, chronic hepatitis B, and history of kidney transplantation, than those with hepatitis A. However, the prevalence of dyslipidemia, malignancy, chronic hepatitis C, and history of liver transplantation did not differ significantly between the groups. For the evaluated risk factors, hepatitis E patients exhibited a significantly higher incidence of consuming undercooked seafood and a history of blood component transfusions than hepatitis A patients (Table 1).

### 3.2. Clinical Manifestations of Hepatitis A and E Infections

The two most common clinical manifestations in this study were jaundice and fever. Hepatitis A patients showed significantly higher proportions of fever (98.0% vs. 38.5%, *p* < 0.001) and jaundice (96.0% vs. 51.9%, *p* < 0.001) than hepatitis E patients. Furthermore, nausea and vomiting were significantly more prevalent in the hepatitis A group than in the hepatitis E group (52.0% vs. 34.6%, *p* = 0.041). In contrast, although diarrhea was a less common clinical manifestation, it occurred more frequently in the hepatitis E group (13.5% vs. 3.0%, *p* = 0.032) than in the hepatitis A group (Table 2).

### 3.3. Laboratory Data

Liver function tests (AST, ALT, alkaline phosphatase, and bilirubin) demonstrated a pattern of hepatocellular injury. Hepatitis A patients exhibited significantly higher peak levels of liver AST and ALT than hepatitis E patients. The total bilirubin peak level in the hepatitis A group was also significantly higher than that in the hepatitis E group. However, hepatitis E patients had significantly lower serum albumin levels than hepatitis A patients (Table 3).

### 3.4. Clinical Predictors to Differentiate Hepatitis E from Hepatitis A

Several clinical predictors were identified that could differentiate hepatitis E patients from hepatitis A patients. In univariate analysis, factors that increased the risk of hepatitis E compared to hepatitis A were age, hypertension, diabetes mellitus, chronic kidney disease, fever, nausea/vomiting, diarrhea, and jaundice. However, after adjusting for the subdistribution of the predictors in multivariable logistic regression analysis, only increased age remained a significant clinical predictor for hepatitis E infection. Conversely, fever and jaundice were more indicative of a hepatitis A infection (Table 4).

### 3.5. Outcomes of Patients Infected with Hepatitis A and E

Table 5 presents the clinical outcomes of hepatitis A and hepatitis E patients, including the duration of transaminitis (determined by ALT levels), duration of jaundice, length of hospital stay, incidence of acute liver failure, and mortality rates. Only the duration of jaundice and length of hospitalization were significantly longer in the hepatitis E group than in the hepatitis A group (43 vs. 26 days, *p* = 0.006 and 7.0 vs. 4.0 days, *p* = 0.001, respectively). Patients infected with HAV tended to exhibit more severe clinical manifestations than hepatitis E patients. Nevertheless, there were no significant differences in hospitalization rates, the incidence of acute liver failure, or mortality rates. Mortality was comparable for the hepatitis A and hepatitis E groups (3.0% vs. 9.6%, *p* = 0.091). Acute liver failure was the cause of death for all patients in this study.

Mortality based on gender showed no significant difference in both the hepatitis A group (3 deaths (5.6%) male vs. 0 deaths (0%) female; *p* = 0.247) and the hepatitis E group (2 deaths (6.9%) male vs. 3 deaths (13%) female; *p* = 0.644). Only 3 (3.1%) patients in the hepatitis A group had HIV infection and none of them experienced death. There were no HIV-infected patients in the hepatitis E group.

Furthermore, the data revealed that 9.6% of HEV-infected patients progressed to chronic hepatitis E. All patients who transitioned to chronic hepatitis E had post-kidney transplantation as an underlying medical condition.

## 4. Discussion

During the 1970s in Thailand, seroprevalence rates for anti-HAV revealed that around 50% of children aged 5–10 years and 100% of 30-year-olds had experienced infection. However, in the 2000s, these rates underwent notable reduction. Specifically, a diminished seroprevalence was noticed among children (less than 5% of 10-year-olds), while young adults exhibited merely 20–30% [13]. The seroprevalence of HAV was also observed to progressively increase with each successive age group [28]. Furthermore, the prevalence of anti-HAV IgG among Thai medical students exhibited a discernible trajectory: 73.01% in 1981, followed by a decrease to 30.23% in 1992, a further decline to 16.67% in 1996, and ultimately reaching 6.67% in 2001. This gradual attenuation in HAV seroprevalence could plausibly be attributed to improved sanitation standards over the intervening years. However, it is noteworthy that the prevalence of anti-HAV IgG in Thai medical students experienced an upswing, ascending to 63.07% in 2012, which might be attributed to the launching of the HAV vaccine in Thailand, which has been available since 1995 [29]. Typically, the HAV vaccine is administered to children from middle- to high-income families during their kindergarten years. Regarding the hepatitis E virus, the prevalence of HEV in Thailand was determined to be 29.7% (95% CI, 26.2–33.4%) in 2019 [30]. In another study among patients undergoing assessment for suspected HEV infection in a tertiary hospital, the prevalence of HEV IgG stood at 65.94% in 2015. Subsequent years showed a decreasing trend: 49.01% in 2016, 37.68% in 2017, and 26.04% in 2018 in the same population. It is notable that the seroprevalence of HEV demonstrated an increase with advancing age groups [31]. The decline in the seroprevalence of both HAV and HEV might be due to improved hygiene in Thailand.

The clinical manifestations, laboratory data, and prognosis of patients with acute viral hepatitis have been well-documented in several studies. Nevertheless, comparative data on acute hepatitis A and hepatitis E are notably lacking, particularly in Thailand, where both acute HAV and HEV cases are sporadic. In developing countries, both HAV and HEV are primarily transmitted via the fecal–oral route [3,4,5,6,9]. However, in Thailand, the clinical symptoms of hepatitis E may differ from those of hepatitis A due to the predominance of zoonotic transmission of hepatitis E genotype 3 [17,18,19]. Consequently, the syndrome data for hepatitis A and hepatitis E may deviate from those reported in other developing countries, including Thailand’s neighboring countries.

This retrospective review examined 152 patients diagnosed with hepatitis A and hepatitis E over 11 years at Siriraj Hospital, Mahidol University. The study provides valuable insights into the clinical features, laboratory data, and clinical outcomes, including mortality. The mean ages of hepatitis A and hepatitis E patients in our study were 32.0 and 46.7 years, respectively. The mean age of the HAV-infected patients in this study aligns with other studies, which reported mean ages ranging from 10 to 30 years [9,32,33,34]. However, the mean age of the HEV-infected patients was higher than that of the hepatitis A group, likely due to the predominance of HEV genotype 3 infection in Thailand. Generally, patients infected with genotype 3 are older than those infected with genotypes 1 and 2. The mean age range for patients infected with genotypes 1 and 2 is 29.3–32.4 years, while the mean for patients infected with genotype 3 is 39.9–64 years [35,36,37,38,39,40,41].

We noted that the hepatitis E group had a higher prevalence of comorbidities, such as hypertension, diabetes mellitus, and chronic kidney disease, than the hepatitis A group. The discrepancy can be explained by the fact that HEV-infected patients who develop symptoms often have some degree of underlying chronic liver disease. Consequently, HEV-infected patients are typically older and present with comorbid diseases. The hepatitis E group also showed a marked increase in post-organ transplant patients, particularly those with post-kidney transplantation. Our study also revealed that only patients with post-kidney transplantation progressed to chronic hepatitis E.

In the analysis of risk factors for virus transmission, our study demonstrated that only the consumption of undercooked seafood was significantly different in the hepatitis E group. Even though HEV genotype 3 is abundant in domestic swine, the consumption of undercooked pork was not a significant factor in the HEV group. The consumption of uncooked seafood is integrated into the popular hotpot dining tradition in Thailand. This practice, characterized by the use of chopsticks, can potentially lead to cross-contamination between raw pork and seafood. Moreover, this could be due to seafood getting contaminated with raw meat, especially pork, during food preparation. Thai culinary culture often involves consuming undercooked seafood, such as marinated seafood (in soy sauce or fish sauce), uncooked oysters, or Thai-style barbecue, which differs from Western cuisine. Contrarily, undercooked pork could also become contaminated with the hepatitis A virus through environmental exposure or during the food preparation process. This could serve as a potential source of transmission via the fecal–oral route, thereby potentially introducing confounding factors into the observations within the hepatitis A group.

The clinical features of hepatitis A and hepatitis E are similar and include fever, jaundice, nausea, vomiting, fatigue, abdominal pain, and myalgia [6,9,12,14,15,21,22,23]. Virtually all of our hepatitis A patients exhibited fever (98%) and jaundice (96%), aligning with previous cohorts reporting prevalence rates of 42–87% for fever [12,21,33] and 62–81% for jaundice [9,12,33]. In the hepatitis E group, the prevalence of jaundice was equal to that of fatigue (51.9%). This finding is similar to the results from a cohort of genotype 3 hepatitis E-infected patients in Italy (jaundice: 33.3–42.8%, fatigue: 42.8–50%) [42]. The maximum levels of total bilirubin and liver enzymes, including AST and ALT, were significantly higher in hepatitis A, but AST and ALT in our study were slightly lower than those in other cohorts. The peak AST, ALT, and total bilirubin ranges from previous studies were 1442–3664 IU/L, 1952–3628 IU/L, and 5.0–13.0 mg/dL, respectively [6,9,21,32,33]. The maximum levels of AST, ALT, and total bilirubin in hepatitis E in a previous study were 178–3608 IU/L, 525–3629 IU/L, and 1.4–19.2 mg/dL [42], and our results align with these values. Consequently, hepatitis A is likely to be more severe than hepatitis E based on both clinical courses and laboratory tests.

This study revealed that patients exhibiting symptoms of fever and jaundice and young patients suspected of having acute viral hepatitis are more likely to be diagnosed with hepatitis A rather than hepatitis E. Although hepatitis A tends to manifest more severe clinical features and laboratory tests, the outcomes, including hospitalization, acute liver failure, and death, were not significantly different. However, the length of hospital stay was higher in the hepatitis E group. This variance may be caused by the longer turnaround time of diagnostic tests for hepatitis E and the fact that the HEV-infected patients were older and had multiple underlying medical conditions.

Hepatitis E infection has shown a clear association with a high mortality rate of up to 41% among pregnant women, particularly during the second and third trimesters [43,44,45]. Furthermore, liver failure is commonly noted in the third trimester, also accompanied by a high mortality rate [43]. However, there were no pregnant women in this study to demonstrate the severity and mortality associated with acute hepatitis E.

Our study has several limitations. First, it was a single-center retrospective review conducted at Siriraj Hospital, located in the capital of Thailand. As such, patient data may not be representative of the broader Thai population or accurately reflect all regions of Thailand with their varied cultures, traditions, and religious practices that may impact dietary habits and hygiene. Moreover, our center is a tertiary university hospital, and the patients may be more severe than those at other hospitals. Second, there was no information on asymptomatic or mildly symptomatic individuals who did not attend the hospital. All data in our study came from ICD diagnostic codes representing symptomatic patients who visited the hospital. Third, because of resource limitations, diagnosing acute HEV infection in this cohort primarily depended on anti-HEV IgM, which exhibits low sensitivity. Consequently, our analysis might potentially underestimate the actual incidence of HEV cases. The utilization of HEV PCR, the current gold standard for HEV diagnosis, would provide a more accurate assessment. Finally, we did not identify or collect data on clinical patterns or extrahepatic symptoms associated with HAV and HEV infections.

## 5. Conclusions

While the clinical courses of acute sporadic hepatitis A and E were similar, the hepatitis A patients almost always presented with fever and jaundice, favoring a hepatitis A diagnosis over hepatitis E. Additionally, the hepatitis A group had higher levels of bilirubin and transaminitis. However, the two groups had no differences in their clinical outcomes, including acute liver failure and mortality.

## Figures and Tables

**Table 1 viruses-15-01888-t001:** Baseline characteristics of patients diagnosed with hepatitis A and E infections.

	Total	Hepatitis A (*n* = 100)	Hepatitis E (*n* = 52)	*p*
Age, years	36.99 ± 15.80	31.95 ± 12.35	46.67 ± 17.26	<0.001
Sex, male, *n* (%)	83 (54.6%)	54 (54%)	29 (55.8%)	0.835
Comorbidity				
Hypertension	13 (8.6%)	4 (4.0%)	9 (17.3%)	0.011
Diabetes mellitus	13 (8.6%)	3 (3.0%)	10 (19.2%)	0.001
Dyslipidemia	12 (7.9%)	5 (5.0%)	7 (13.5%)	0.109
Chronic kidney disease	10 (6.6%)	1 (1.0%)	9 (17.3%)	<0.001
Malignancy	6 (3.9%)	2 (2.0%)	4 (7.7%)	0.181
Underlying liver diseases				
Chronic hepatitis B	5 (3.3%)	1 (1.0%)	4 (7.7%)	0.047
Chronic hepatitis C	2 (1.3%)	0 (0%)	2 (3.8%)	0.116
Cirrhosis	4 (2.6%)	1 (1.0%)	3 (5.8%)	0.116
Post-organ transplantation				
Kidney transplantation	8 (5.3%)	0 (0%)	8 (15.4%)	<0.001
Liver transplantation	2 (1.3%)	0 (0%)	2 (3.8%)	0.116
Risk factors				
History of blood transfusion	3 (2.0%)	0 (0%)	3 (5.8%)	0.039
PWID	0 (0%)	0 (0%)	0 (0%)	N/A
Undercooked pork consumption	6 (3.9%)	2 (2.0%)	4 (7.7%)	0.181
Undercooked seafood consumption	4 (2.6%)	0 (0%)	4 (7.7%)	0.013

Data are expressed as the means ± SDs or numbers (percentages). Abbreviation: PWID, people who inject drugs; N/A, not applicable.

**Table 2 viruses-15-01888-t002:** Clinical characteristics of patients with acute viral hepatitis A and E.

Symptoms	Total	Hepatitis A (*n* = 100)	Hepatitis E (*n* = 52)	*p*
Fever	118 (77.6%)	98 (98.0%)	20 (38.5%)	<0.001
Loss of appetite	67 (44.1%)	46 (46.0%)	21 (40.4%)	0.508
Myalgia	34 (22.4%)	26 (26.0%)	8 (15.4%)	0.136
Fatigue	70 (46.1%)	43 (43.0%)	27 (51.9%)	0.295
Nausea/vomiting	70 (46.1%)	52 (52.0%)	18 (34.6%)	0.041
Diarrhea	10 (6.6%)	3 (3.0%)	7 (13.5%)	0.032
Abdominal pain	41 (27.0%)	32 (32.0%)	9 (17.3%)	0.053
Jaundice	123 (80.9%)	96 (96.0%)	27 (51.9%)	<0.001

Data are expressed as numbers (percentages).

**Table 3 viruses-15-01888-t003:** Laboratory data of patients with acute viral hepatitis A and E.

Laboratory	Total	Hepatitis A (*n* = 100)	Hepatitis E (*n* = 52)	*p*
Maximum AST level unit/L	760 (224–1553)	1004 (308–1654)	443 (199–1122)	0.014
Maximum ALT level unit/L	1003 (403–1950)	1551 (733–2320)	539 (263–1285)	<0.001
Maximum ALP level unit/L	192.5 (145–249)	193 (154–248.5)	183.5 (126–267)	0.699
Maximum total bilirubin level mg/dL	8.60 (4.80–14.90)	8.94 (6.31–13.34)	5.90 (1.16–20.81)	0.030
Lowest albumin level g/dL	3.70 (3.20–4.10)	3.80 (3.50–4.20)	3.45 (2.90–4.00)	0.013
Highest prolonged INR	1.21 (1.14–1.52)	1.18 (1.14–1.43)	1.30 (1.15–1.68)	0.272

Data are expressed as medians (interquartile ranges). Abbreviations: ALP, alkaline phosphatase; ALT, alanine aminotransferase; AST, aspartate aminotransferase; INR, international normalized ratio.

**Table 4 viruses-15-01888-t004:** Univariate and multivariate analyses of clinical predictors differentiating hepatitis E from hepatitis A infections.

	Univariable Analysis Crude OR (95% CI)	Crude *p*	Multivariable Analysis Adjusted OR (95% CI)	*p*
Age (years)	1.07 (1.04–1.10)	<0.001	1.06 (1.01–1.13)	0.035
Hypertension	5.02 (1.47–17.21)	0.010		
Diabetes mellitus	7.70 (2.02–29.40)	0.003		
Chronic kidney disease	20.72 (2.55–168.66)	0.005		
Chronic hepatitis B	8.25 (0.90–75.83)	0.062		
Fever	0.01 (0.003–0.06)	<0.001	0.03 (0.01–0.29)	0.002
Nausea and vomiting	0.49 (0.24–0.98)	0.043		
Diarrhea	5.03 (1.24–20.35)	0.024		
Jaundice	0.05 (0.01–0.14)	<0.001	0.05 (0.01–0.27)	0.001

**Table 5 viruses-15-01888-t005:** Outcome comparison of patients with hepatitis A and E infections.

	Total	Hepatitis A (*n* = 100)	Hepatitis E (*n* = 52)	*p*
Duration of transaminitis, ALT (days)	43 (28–78)	35 (26–80)	47 (28–74)	0.595
Duration of jaundice (days)	32 (21–57)	26 (20–37)	43 (22–79)	0.006
Hospitalization	79 (52.0%)	47 (47.0%)	32 (61.5%)	0.089
Duration of hospitalization (days)	5.0 (3.0–7.0)	4.0 (2.0–5.0)	7.0 (4.0–13.0)	0.001
Acute liver failure	12 (7.9%)	5 (5.0%)	7 (13.5%)	0.109
Death	8 (5.3%)	3 (3.0%)	5 (9.6%)	0.091

Data are expressed as medians (interquartile ranges) or numbers (percentages).

## Data Availability

Not applicable.

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
