# Peer review of "A Comparison of Clinical Manifestations and Outcomes between Acute Sporadic Hepatitis A and Hepatitis E Infections in Thailand"

_viruses, 2023, doi:10.3390/v15091888_

Round 1

Reviewer 1 Report

Khongviwatsathien et al; compared the clinical manifestations and outcomes between acute sporadic Hepatitis A and Hepatitis E infections in Thailand. The study is a retrospective study for 152 patients diagnosed with acute hepatitis A or hepatitis E from January 2007 to August 2018 at Siriraj Hospital in Thailand. The study conclude that the clinical manifestations of hepatitis A and hepatitis E were comparable. However, fever and jaundice were more prevalent and aminotransferase and bilirubin levels were higher in the HAV- infected group. There are some limitations of the study due to the limited number of patients enrolled over a 11 years, and the site of the hospital in which the patients were selected. 

Additionally, the authors need to address the following concern:

1.     What the gender of the patients and did the gender affects the mortality especially in HEV infection.

2.     What is the sensitivity and the specificity of the assays used for the diagnosis of HEV and HAV.

3.     How they did typing for HEV since they did not do PCR for each subjects.

4.     Are any of the patients HIV + ve and how it affects the morbidity of the infection.

5.     How to explain the significance of the consumption of undercooked seafood in HEV if it is genotype 3 infection (and not undercooked pork).

6.     What is the prevalence of anti-HEV IgG and anti-HAV IgG in comparable control age groups at the same site?

Appropriate

Author Response

  • What the gender of the patients and did the gender affects the mortality especially in HEV infection.

Mortality based on gender showed no significant difference in both the hepatitis A group (3 deaths (5.6%) male vs 0 deaths (0%) vs female; p = 0.247) and the hepatitis E group (2 deaths (6.9%) male vs 3 deaths (13%) female; p = 0.644).

This information has been added to the manuscript.

  • What is the sensitivity and the specificity of the assays used for the diagnosis of HEV and HAV.

The sensitivity and specificity of HAV IgM following the manufacturer's instructions were 98.58% and 99.0%, respectively. However, the post-marketing data revealed sensitivity and specificity of HAV IgM were 78.2-89.3% and 100%, respectively.

The sensitivity and specificity of HEV IgM following the manufacturer's information were 100% and 100%, respectively. However, subsequent post-marketing data indicated a sensitivity range of 24% to 61.5% and a consistent specificity of 100% for HEV IgM.

This information has been added to the manuscript.

  • How they did typing for HEV since they did not do PCR for each subjects.

Because HEV viremia is transient and no commercial HEV PCR test is currently available, compounded by many patients presenting during the later stages of the clinical course. Only 14 patients (26.9%) in the hepatitis E group were confirmed with HEV PCR for diagnosis. However, patients showing acute hepatitis symptoms with positive anti-HEV IgM, which has 100% specificity, might indicate compatibility with acute hepatitis E infection.

  • Are any of the patients HIV + ve and how it affects the morbidity of the infection.

Only 3 (3.1%) patients in the hepatitis A group had HIV infection and none of them experienced death. There were no HIV-infected patients in the hepatitis E group.

This information has been added to the manuscript.

  • How to explain the significance of the consumption of undercooked seafood in HEV if it is genotype 3 infection (and not undercooked pork).

The consumption of uncooked seafood is integrated into the popular hotpot dining tradition in Thailand. This practice, characterized by the use of chopsticks, can potentially lead to cross-contamination between raw pork and seafood. Moreover, this could be due to seafood getting contaminated with raw meat, especially pork, during food preparation. Thai culinary culture often involves consuming undercooked seafood, such as marinated seafood (in soy sauce or fish sauce), uncooked oysters, or Thai-style barbecue, which differs from Western cuisine. Contrarily, undercooked pork could also become contaminated with the hepatitis A virus through environmental exposure or during the food preparation process. This could serve as a potential source of transmission via the fecal-oral route, thereby potentially introducing confounding factor into the observations within the hepatitis A group.

The discussion about the prevalence rate of HAV and HEV in Thai population has been added to the manuscript.

  • What is the prevalence of anti-HEV IgG and anti-HAV IgG in comparable control age groups at the same site?

During the 1970s in Thailand, seroprevalence rates for anti-HAV revealed that around 50% of children aged 5–10 years and one hundred percent of 30-year-olds had experienced infection. However, as the 2000s, these rates underwent notable reduction. Specifically, a diminished seroprevalence was noticed among children (less than 5% of 10-year-olds), while young adults exhibited merely 20%–30%. The seroprevalence of HAV was also observed to progressively increase with each successive age group. Furthermore, the prevalence of anti-HAV IgG among Thai medical students exhibited a discernible trajectory: 73.01% in 1981, followed by a decrease to 30.23% in 1992, a further decline to 16.67% in 1996, and ultimately reaching 6.67% in 2001. This gradual attenuation in HAV seroprevalence could plausibly be attributed to improved sanitation standards over the intervening years. However, it is noteworthy that the prevalence of anti-HAV IgG in Thai medical students experienced an upswing, ascending to 63.07% in 2012, which might be attributed to the launching of the HAV vaccine in Thailand, which has been available since 1995. Typically, the HAV vaccine is administered to children from middle to high-income families during their kindergarten years. Regarding the hepatitis E virus, the prevalence of HEV in Thailand was determined to be 29.7% (95% CI, 26.2-33.4%) in 2019. In another study among patients undergoing assessment for suspected HEV infection in a tertiary hospital, the prevalence of HEV IgG stood at 65.94% in 2015. Subsequent years showed a decreasing trend: 49.01% in 2016, 37.68% in 2017, and 26.04% in 2018 in the same population. It's notable that the seroprevalence of HEV demonstrated increasing with advancing age groups. The decline in the seroprevalence of both HAV and HEV might be an effect from an improved hygiene in Thailand.

The discussion about the prevalence rate of HAV and HEV in Thai population has been added to the manuscript.

Reviewer 2 Report

This manuscript “A Comparison of Clinical Manifestations and Outcomes Between Acute Sporadic Hepatitis A and Hepatitis E Infections in Thailand” by Khongviwatsathien et al describes the clinical manifestations, laboratory findings, and outcomes of hepatitis A and hepatitis E infections in Thailand. This manuscript is well written.

Comments:

1.       Please include data on HAV and HEV antibody titers, and HEV-PCR.

2.       What is HAV-HEV co-infection status within the study population?

3.       Including samples from recent years, i.e. 2020-2023 would be interesting.

4.       Discussion section needs streamlining. Lines from introduction and results are repeated in discussion section, these can be removed.

5.       Discussing current prevalence rate of HAV and HEV in human population in Thailand would be essential. This would enable the readers to understand why understanding differentiating clinical manifestation of HAV and HEV is important.

Redundancy: Lines from introduction and results are repeated in discussion. 

Author Response

  • Please include data on HAV and HEV antibody titers, and HEV-PCR.

Based on the IgM detection techniques, there was no antibody titer in both HAV and HEV antibody detections. In our lab, the reports showed only serum cut-off values that were interpreted as positive, negative, or borderline.

However, The HEV PCR data has been added to the manuscript.

  • What is HAV-HEV co-infection status within the study population?

There was no hepatitis A and hepatitis E co-infection in this study.

This information has been added to the manuscript.

  • Including samples from recent years, i.e. 2020-2023 would be interesting.

This study has concluded, so the further data are not available.

  • Discussing current prevalence rate of HAV and HEV in human population in Thailand would be essential. This would enable the readers to understand why understanding differentiating clinical manifestation of HAV and HEV is important.

During the 1970s in Thailand, seroprevalence rates for anti-HAV revealed that around 50% of children aged 5–10 years and one hundred percent of 30-year-olds had experienced infection. However, as the 2000s, these rates underwent notable reduction. Specifically, a diminished seroprevalence was noticed among children (less than 5% of 10-year-olds), while young adults exhibited merely 20%–30%. The seroprevalence of HAV was also observed to progressively increase with each successive age group. Furthermore, the prevalence of anti-HAV IgG among Thai medical students exhibited a discernible trajectory: 73.01% in 1981, followed by a decrease to 30.23% in 1992, a further decline to 16.67% in 1996, and ultimately reaching 6.67% in 2001. This gradual attenuation in HAV seroprevalence could plausibly be attributed to improved sanitation standards over the intervening years. However, it is noteworthy that the prevalence of anti-HAV IgG in Thai medical students experienced an upswing, ascending to 63.07% in 2012, which might be attributed to the launching of the HAV vaccine in Thailand, which has been available since 1995. Typically, the HAV vaccine is administered to children from middle to high-income families during their kindergarten years. Regarding the hepatitis E virus, the prevalence of HEV in Thailand was determined to be 29.7% (95% CI, 26.2-33.4%) in 2019. In another study among patients undergoing assessment for suspected HEV infection in a tertiary hospital, the prevalence of HEV IgG stood at 65.94% in 2015. Subsequent years showed a decreasing trend: 49.01% in 2016, 37.68% in 2017, and 26.04% in 2018 in the same population. It's notable that the seroprevalence of HEV demonstrated increasing with advancing age groups. The decline in the seroprevalence of both HAV and HEV might be an effect from an improved hygiene in Thailand.

The discussion about the prevalence rate of HAV and HEV in Thai population has been added to the manuscript.

Round 2

Reviewer 1 Report

The authors responded adequately to my comments